# Metabolomics Characterize the Differential Metabolic Markers between Bama Xiang Pig and Debao Pig to Identify Pork

**DOI:** 10.3390/foods12010005

**Published:** 2022-12-20

**Authors:** Changyi Chen, Junwen Zheng, Chenyong Xiong, Hongjin Zhou, Chuntao Wei, Xin Hu, Xinxiu Qian, Mengyi He, Yandi Shi, Yuwen Liu, Zongqiang Li

**Affiliations:** 1College of Animal Science and Technology, Guangxi University, Nanning 530003, China; 2Shenzhen Branch, Guangdong Laboratory of Lingnan Modern Agriculture, Key Laboratory of Livestock and Poultry Multi-Omics of MARA, Agricultural Genomics Institute at Shenzhen, Chinese Academy of Agricultural Sciences, Shenzhen 518124, China

**Keywords:** Bama pig, differential metabolites, identification, meat quality, LC-MS

## Abstract

The Bama Xiang pig (BM) is a unique pig species in Guangxi Province, China. Compared to other breeds of domestic pig, such as the Debao pig (DB), it is smaller in size, better in meat quality, resistant to rough feeding and strong in stress resistance. These unique advantages of Bama Xiang pigs make them of great edible value and scientific research value. However, the differences in muscle metabolites between Bama Xiang pigs (BM) and Debao pigs (DB) are largely unexplored. Here, we identified 214 differential metabolites between these two pig breeds by LC-MS. Forty-one such metabolites are enriched into metabolic pathways, and these metabolites correspond to 11 metabolic pathways with significant differences. In Bama pigs, the abundance of various metabolites such as creatine, citric acid, L-valine and hypoxanthine is significantly higher than in Debao pigs, while the abundance of other metabolites, such as carnosine, is significantly lower. Among these, we propose six differential metabolites: L-proline, citric acid, ribose 1-phosphate, L-valine, creatine, and L-arginine, as well as four potential differential metabolites (without the KEGG pathway), alanyl-histidine, inosine 2′-phosphate, oleoylcarnitine, and histidinyl hydroxyproline, as features for evaluating the meat quality of Bama pigs and for differentiating pork from Bama pigs and Debao pigs. This study provides a proof-of-concept example of distinguishing pork from different pig breeds at the metabolite level and sheds light on elucidating the biological processes underlying meat quality differences. Our pork metabolites data are also of great value to the genomics breeding community in meat quality improvement.

## 1. Introduction

In recent years, there have been many aspects of research on pork quality. Driessen and others said in their research articles that fasting before slaughter can affect meat quality by increasing pH value [1]. Acevedo Giraldo’s study also described the impact of premortem fasting on meat quality, and they suggested in their article that the total fasting time should not exceed 12 h [2]. In addition, there are many studies on the impact of different treatments before slaughter on pork quality [3,4,5,6]. It is not just the treatment before slaughter that affects the meat quality of pork, but also the breeding environment and feeding conditions during the fattening period [7,8,9]. In addition, there are many other factors that will affect the meat quality of pork, such as the residual feed intake [10,11,12]. Ngapo and others summarized the factors that affect the meat quality of pork [13]. Because of the diversity of influencing factors, we are required to evaluate the quality of pork. In the past, we generally used meat quality traits to evaluate and describe pork, describing pork meat quality in terms of external traits such as pH, meat color, dripping loss and shear force [14,15,16]. However, with the development of science and technology, it is not enough to use these objective indicators to evaluate pork quality. Many new tests and indicators are used to evaluate pork meat quality, such as proteomics, developed in recent years [17]. Since the quality properties of meat are greatly affected by the environment, biomarkers are better than genetic markers for evaluating meat quality [18]. Some researchers have also successfully screened some biomarkers that can be used for meat quality evaluation. Due to the fact that some of the amino acids can provide pork flavor [19], which can improve the flavor of meat, numerous studies also put forward methods to improve meat quality based on this [20,21,22,23]. Similar metabolomics methods were used in this study. Through non-targeted metabolomics, some metabolites related to pork meat quality traits were preliminarily selected as biomarkers for the evaluation of pork meat quality. Some people have used this method to search for biomarkers previously. Yu looked for biomarkers based on the research of the impact of aging on meat quality through metabolomics [24], and Cao looked for biomarkers to distinguish dead and live pork through the combination of non-targeted metabolomics and pseudo-targeting [25]. Shuji Ueda [26] identified metabolites in livestock meat through metabolomics, which ca be used as meat quality evaluation indicators, and even used to distinguish cattle breeds.

Compared with other domestic pig breeds, the Bama Xiang pig has its own unique advantages. The Bama Xiang pig (BM) is smaller, but it has the benefits of high meat quality, rough feeding resistance, strong stress resistance and others. Many studies have shown these advantages for the BM breed. In addition, BM is also used in numerous studies, and in recent years, BM has been used for organ transplantation research. Owing to the physical condition of BM, after knocking out the immune rejection-related genes, it can be well matched with the human body to realize organ transplantation. BM has better meat quality than other pigs, which means it has a higher edible value. This study aims to compare the metabolites of the dorsal longest muscle of BM and Debao pig (DB) by LC-MS, understand the interaction of these metabolites and related metabolic pathways, screen the differential metabolism that can be used to identify these two kinds of domestic pigs, provide a theoretical basis for the breed breeding and meat quality identification of BM, and pave the way for further exploration of the biological causes of the formation of BM pork quality.

## 2. Materials and Methods 

### 2.1. Materials

Bama pigs and Debao pigs were purchased from the original breeding farm in Bama County, Guangxi, after fresh slaughter. The breeding conditions of these pigs are the same. They are three months old, castrated boars. Both were sampled after slaughter. We sampled the longissimus dorsi after the slaughter at 5 am. The samples were put immediately into liquid nitrogen for quick freezing for 15 min, then taken out and put into dry ice and transported to the laboratory, and then the experiment was started in the laboratory. The reagents used were methanol, formic acid, water and acetonitrile, which were purchased from Thermo company, and l-2-chlorophenyl alanine was purchased from Shanghai Hengchuang Biotechnology Co., Ltd. All chemicals and solvents are analytical grade or chromatographic grade. The instruments used were: an automatic sample rapid grinder (Wonbio-E Shanghai Wanbo Biotechnology Co., Ltd. Shanghai, China), an ultrasonic cleaner (F-060SD Shenzhen Fuyang Technology Group Co., Ltd. Shenzhen, China), a desktop high-speed refrigerated centrifuge (TGLl-16MS Shanghai Luxiangyi Centrifuge Instrument Co., Ltd. Shanghai, China), a freeze concentration centrifugal dryer (LNG-T98 Taicang Huamei Biochemical Instrument Factory, Suzhou, China), a high-resolution mass spectrometer (QE Thermofisher Technology Company, Waltham, Massachusetts, USA), a high-performance liquid chromatograph (Nexera UPLC Shimadzu Company of Japan, Kyoto, Japan), and a chromatographic column (ACQUITY UPLC HSS T3 (100 mm × 2.1 mm, 1.8 um) Waters).

### 2.2. Detection of Metabolites by LC-MS

We accurately weighed 30 mg of longissimus dorsi into a 1.5 mL EP tube and added an internal standard (l-2-chlorophenyl alanine, 0.06 mg/mL; methanol configuration) 20 μL. Joined 400 μL methanol–water (*v*:*v* = 4:1);We added two small steel balls, pre-cooling them in the refrigerator at −20 °C for 2 min, and then putting them into the grinder for grinding (60 Hz, 2 min);The next step was ultrasonic extraction in an ice water bath for 10 min, stood at −20 °C for 30 min;The sample was centrifuged for 10 min (13,000 rpm, 4 °C), 300 μL supernatant was put into the LC-MS injection vial and evaporated;Then 300 μL methanol–water (*v*:*v* = 1:4) was used for re-dissolution (vortexed for 30 s, ultrasound for 3 min); stood for 2 h at −20 °C;The sample was centrifuged for 10 min (13,000 rpm, 4 °C), and 150 μL of supernatant was removed with a syringe, then using a 0.22 μm organic phase pinhole filter, the supernatant was transferred to an LC injection vial and stored at −80 °C until LC-MS analysis.Quality control samples (QC) were prepared by mixing an extract of all the samples in equal volumes.

(all extraction reagents were pre-cooled at −20 °C before use.)

Conditions for LC-MS:

Chromatographic column: ACQUITY UPLC HSS T3 (100 mm × 2.1 mm, 1.8 um);

Column temperature: 45 °C;

Mobile phase: A-water (containing 0.1% formic acid), B-acetonitrile (containing 0.1% formic acid); Flow rate: 0.35 mL/min;

Injection volume: 2 μL.

The sample mass spectrum signals were collected by positive and negative ion scanning modes.

### 2.3. Data Analysis Methods

The original data obtained by LC-MS were processed by metabolomics processing software Progenesis QI v2.3 (Nonlinear Dynamics, Newcastle, UK) for baseline filtering, peak recognition, integration, retention time correction, peak alignment and normalization. Multivariate statistical analyses such as PCA, PLS-DA, and OPLS-DA were carried out on the processed data (Appendix A). Principal component analysis (PCA, Figure 1a) of metabolites could reflect the variability between and within sample groups, observe the overall distribution trend between samples, and judge the possible discrete points. The main parameter of the PCA model is R2X. The abscissa t [1] and ordinate t [2] of the PCA score map represent the score values projected on the principal components PC1 and PC2 of each sample, respectively. The projection score of each sample on the plane is composed of the first principal component and the second principal component, the spatial coordinate, which can intuitively reflect the similarity or difference between samples. If the difference between the two samples is significant, the two coordinate points are relatively far away on the score chart and vice versa. The prediction of sample categories was performed by PLS-DA (Figure 1b). Adding grouping variables to PLS-DA can make up for the shortcomings of the PCA method, and the parameter R2X (cum) can evaluate the effectiveness of the model. In addition to the parameter R2X (cum), PLS-DA also includes the interpretation rate R2Y (cum) and the prediction rate Q2 (cum). The closer the two are to one, the better the PLS-DA model can explain and predict the difference between the two groups of samples, representing the better prediction ability of the model (Figure 1d). OPLS-DA (Figure 1c) was used to filter out the noise irrelevant to the classification information, improve the analytical ability and effectiveness of the model, and maximize the differences between different groups within the model. The OPLA-DA score chart has two principal components: the predictive principal component and the orthogonal principal component. There is only one predictive principal component, while there can be multiple orthogonal principal components. OPLS-DA maximizes the difference between groups on t1, so it can directly distinguish the variation between groups from t1, while the orthogonal principal component reflects the variation within groups. There are significant differences between the two groups on the OPLS-DA score chart. In the loading OPLS DA diagram (Figure 1e), the closer the metabolic ions are to −1 and 1, the greater the concerns. The metabolites closer to the two corners in the Splot-OPLS-DA diagram (Figure 1f) are also important. Then univariate analysis was carried out to obtain the volcano map of metabolites. Finally, the metabolites were screened to obtain differential metabolites. Correlation analysis was carried out on these differential metabolites, and finally, the metabolic pathway and network diagram were analyzed.

## 3. Results 

### 3.1. Multivariate Statistical Analysis and Univariate Statistical Analysis

For the PCA (Figure 1a), the contribution rate of PC1 was 36.1%; the contribution rate of PC2 was 14.9%, and the cumulative contribution rate was 51%. For the PLS-DA (Figure 1b), the contribution rate of PC1 was 41.9%; the contribution rate of PC2 was 6.23%, and the cumulative contribution rate was 48.13%. For the OPLS-DA (Figure 1c), the contribution rate of PC1 was 40.6%; the contribution rate of PC2 was 7.35%, and the cumulative contribution rate was 47.95%. Colored ellipses represented 95% confidence intervals. The results show that the two groups of samples were gathered into two clusters. Through those analyses (Figure 1d–f), we obtained some important metabolites, such as carnosine, orciprenaline, PC (16:0/20:3(8Z,11Z,14Z)), and PC (18:1(11Z)/16:0). These metabolites were also screened out as differential metabolites by other methods later.

Through univariate statistical analysis of the data matrix, the volcanic map (Figure 2) showed the distribution of different metabolites in the two kinds of pigs. Red indicated that the content of metabolites in BM was significantly higher than that in DB, blue indicated that it was significantly less than that in DB, and gray indicated that the content of metabolites in the two kinds of pigs was not significantly different. These points represented different metabolites. 

### 3.2. Differential Metabolites and Correlation Analysis

Next, the differential metabolites were screened. The screening conditions were VIP > 1 and *p* < 0.01. A total of 213 differential metabolites of BM and DB were obtained (Appendix A). From the differential metabolite heat map (Figure 3), we know the content of differential metabolites in each sample detected. The color from blue to red indicated that the expression abundance of metabolites was from low to high. There was a significant difference in metabolite levels between BM and DB (Figure 3). The darker the red, the higher the abundance of differential metabolites. There were obvious differences between BM and DB. The correlation analysis chart (Figure 4) showed the degree of correlation between the two different metabolites. Red indicates a positive correlation, and the darker the color, the greater the correlation.

We found that some metabolites in BM and DB had highly significant differences, such as PC (18:1 (11Z)/0:0), PE (14:0/24:0), PE (18:0/0:0) and PC (18:1 (11Z)/16:0) and other lipids and lipoid molecules; creatine, L-valine, alanyl-glutamine and other organic acids and their derivatives. These were preliminarily screened differential metabolites that could be used as biomarkers. Through the correlation analysis of metabolites, we can know that among the lipids and lipid molecules such as PC (16:0/00:0) have a high correlation, while they have a negative correlation with stearoylcarnitine and other fatty acyls. Among the fatty acyls like stearoylcarnitine that show a positive correlation, L-valine has a negative correlation with N-acetyl-L-histidine and carnosine and a positive correlation with citric acid. All these are organic acids and their derivatives. 

Therefore, the clustering heat map (Figure 3) of different metabolites accurately screened the metabolites with significant differences in the pork samples. The two groups of samples were clustered together by comparing the different metabolites of the two pig muscles. The two kinds of pigs were distinguished according to the type and content of different metabolites in their muscles.

### 3.3. Path Analysis and Network Diagram

The KEGG database was used for the pathway enrichment of metabolites. We only focused on the analysis of differential metabolites that can be enriched into the metabolic pathway. There are 41 differential metabolites that can be enriched in the metabolic pathway, of which 26 are upregulated in BM (Figure 5a and Table 1), and 15 are downregulated (Figure 5b and Table 2). According to classification statistics, more than half of the metabolites are organic acids, such as amino acids (3-methyl-l-histidine, 4-hydroxyproline and their derivatives). In the aminoacyl-tRNA biosynthesis pathway of BM, the expression of histidine was downregulated, and the expression of arginine, glutamine, lysine, proline and valine was increased; In the ABC transporters pathway, as in the previous pathway, the expression of 4-hydroxyproline and taurine were also upregulated. We calculated statistics on differential metabolites. The following figure shows the classification of metabolites that could be enriched to the KEGG pathway. In addition, we also made statistics on other differential metabolites, mainly lipids and lipid molecules, organic acids and their derivatives, as well as some unclassified metabolites. Through the pathway enrichment diagram (Figure 6) and bubble diagram (Figure 7), we found 11 signal pathways with highly significant differences. Most of the other metabolic pathways are related to the biosynthesis of various amino acids, such as glycine, threonine, and leucine. We noticed that there were 11 signal pathways with significant differences. Finally, we used the network diagram to make a network diagram of metabolic pathways and corresponding metabolites (Figure 8), which can further clarify the relationship between different metabolites and metabolic pathways. 

## 4. Discussion

Metabolomics has become a new method for analyzing the chemical makeup of organisms in recent years and has a broad range of applications. For example, metabolomics can be used to explore the mechanisms of action for various pathogens, identify biomarkers that can be used to predict the occurrence of disease, and monitor the course of the disease. Some studies have also shown that metabolomics can be used to analyze the biological processes of disease pathogenesis and provide a basis for disease treatment. In this study, metabolomics was used to explore the metabolite differences between BM and DB, and then biomarkers were identified that could be used to grade the meat quality of BM. Furthermore, the key metabolites of pork meat quality formation and the biological processes of these molecules were explored. Potential metabolites that can be studied further to identify different pork varieties have been reported. We found a total of 26 upregulated metabolites in BM that may be related to meat quality formation and could act as biomarkers for the meat quality rating (Table 1). These metabolites are basically organic acids and their derivatives, lipids, nucleotides and organic oxygen compounds (Figure 5a). When we annotated these metabolites, we found that many of them were linked to the metabolic processes of amino acids and purines. The network diagram (Figure 8) and metabolic pathway enrichment analysis clarified these relationships, showing that 3-methyl-L-histidine is involved in histidine biosynthesis, γ-glutamyl-L-putrescine in arginine and proline metabolism, and creatine in arginine, serine, proline, glycine, and threonine metabolism. Overall, these metabolites are usually responsible for meat quality. Moreover, metabolic pathways with significant differences were also screened, and most of them are associated with the digestion, absorption, and metabolism of proteins, amino acids, and lipids. Most of these metabolic pathways are associated with the digestion, absorption, and metabolism of proteins, amino acids, and lipids. In the metabolic pathway of protein digestion and absorption, endogenous and exogenous proteins are digested by various enzymes into various amino acids in the body. Among these amino acids are L-proline, glycine, β-alanine, and L-lysine. In the aminoacyl-tRNA biosynthesis pathway, the levels of L-glutamic acid, valine, L-lysine, L-proline and L-arginine were higher than the average level, and the content of L-histidine was lower than average. L-lysine and other organic acids and their derivatives are involved in the metabolic pathways of amino acid biosynthesis, protein digestion and absorption; N-acetylamine plays an important role in arginine biosynthesis by promoting the urea cycle [27]. The role of L-glutamine and citric acid in the metabolic pathways of alanine, aspartate, and glutamate metabolism is to promote the biosynthesis of glycine, threonine and other amino acids, which is consistent with previous research [28,29]. Glutamine can lead to protein synthesis and reduced muscle catabolism [30,31]. Altogether these are the major amino acids involved in meat formation. Citric acid plays an important role in the tricarboxylic acid cycle. Some of the detected amino acids come from daily diets, such as feed, and have a profound impact on the quality of meat. Therefore, in addition to the problems of pig breeds, feed ingredients also have a certain influence on pork; however, in our study, no changes were made to the feed content. Significantly downregulated metabolites (Table 2) also play an important role. For example, creatine can be converted into phosphocreatine to become a ready-made source of ATP [31], and 2-hydrocinnamic acid exhibits antioxidant properties [32]. The amino acid contents of the two pig types were compared and analyzed. The results showed that the content of flavor amino acids (aspartate, glycine, aspartate, alanine, isoleucine, proline, and serine) [33] and essential amino acids (lysine, methionine, threonine, leucine, isoleucine, valine, and phenylalanine) in BM were higher than those in DB. This directly shows that BM has better pork quality and flavor. 

In addition to amino acids and their derivatives, we also detected the presence of other metabolites, such as lipids and organic oxygen compounds. The organic oxygen compounds, L-fucose 1-phosphate, ribose 1-phosphate, and uridine diphosphate N-acetylglucosamine, are involved in the metabolism of guanosine diphosphate, a key substrate for carbohydrate and sugar ester synthesis. L-fucose 1-phosphate also showed significant differences and plays a role in the metabolic pathways of amino sugar and nucleoside sugars (Figure 8). Furthermore, we also screened several potential metabolites that can be used for meat quality evaluation from those metabolites that have not been enriched in metabolic pathways, such as alanyl-histidine, inosine 2′-phosphate, oleoylcarnitine, and histidine-hydroxyproline. Alanyl-histidine is a dipeptide carnosine, an effective anti-saccharifying and antioxidant agent [34,35]; carnosine protects cell telomeres from damage [36]. These metabolites play important roles in meat production. Further statistical analysis of the potential differential metabolites was performed to identify those that could be significant biomarkers for pork identification. We screened six metabolites: L-proline, citric acid, ribose 1-phosphate, L-valine, creatine and L-arginine, to identify BM and DB breeds. Four potential differential metabolites, alanyl-histidine, inosine 2′-phosphate, oleoylcarnitine and histidine-hydroxyline, were used as the identification metabolites for the pigs. 

In the follow-up experiments, we will explore the biological processes of meat quality formation by analyzing the metabolic pathways of metabolites related to this process. Moreover, metabolites that have not been annotated require further investigation.

## 5. Conclusions

The research and analysis of results showed that there are obvious differences in the metabolites in the main metabolic pathways of the longissimus muscle of BM and DB. We found 214 different metabolites and then preliminarily screened six to identify the pigs. These consisted of L-proline, citric acid, ribose 1-phosphate, L-valine, creatine and L-arginine. The four potential differential metabolites, alanyl histidine, inosine 2′-phosphate, oleoylcarnitine and histidine–hydroxyline, were used as the identification metabolites for BM and DB pigs. Significant differences were observed in the screened metabolites between the two types of pork, indicating that they can be used to identify BM and DB. This study details a basic research method for the identification of many kinds of domestic pigs and may aid in the detection of more varieties of pork. Overall, this study data can provide a basis for improved pork safety and traceability in the future.

## Figures and Tables

**Figure 1 foods-12-00005-f001:**
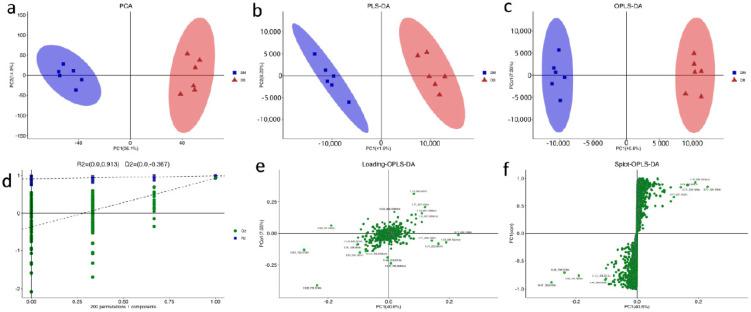
Statistical analysis of metabolites between BM and DB. Principal component analysis (PCA) results (**a**) PLS-DA analysis result (**b**) OPLA-DA analysis result (**c**) showed that the clustering of each group is strong, with little difference within the group, but the difference between groups is obvious. All of these models were evaluated using parameter R2X (cum) (**d**). We used the Loading OPLA-DA to find the most important metabolic ions close to −1 and 1 (**e**). We used the Splot-OPLS-DA to screen the metabolites closer to the two corners (**f**). BM: Bama xiang pig; DB: Debao pig.

**Figure 2 foods-12-00005-f002:**
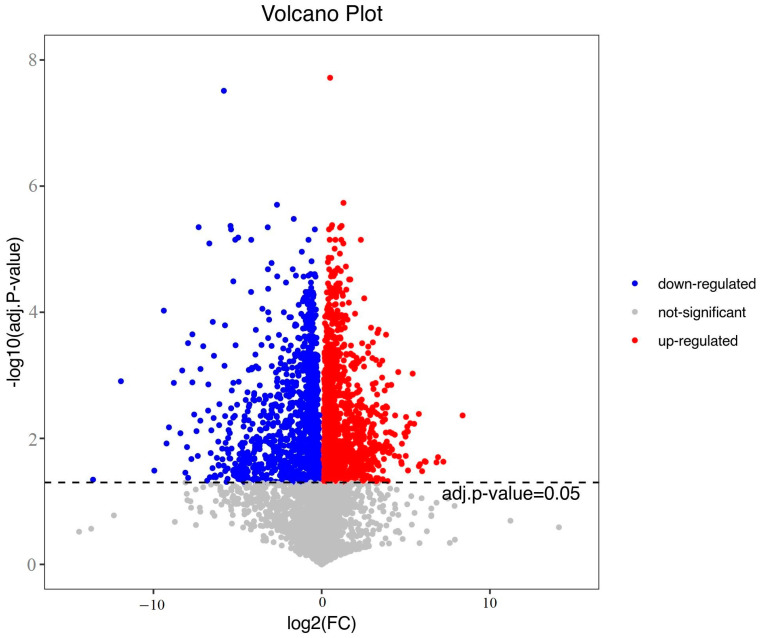
Univariate statistical analysis result.

**Figure 3 foods-12-00005-f003:**
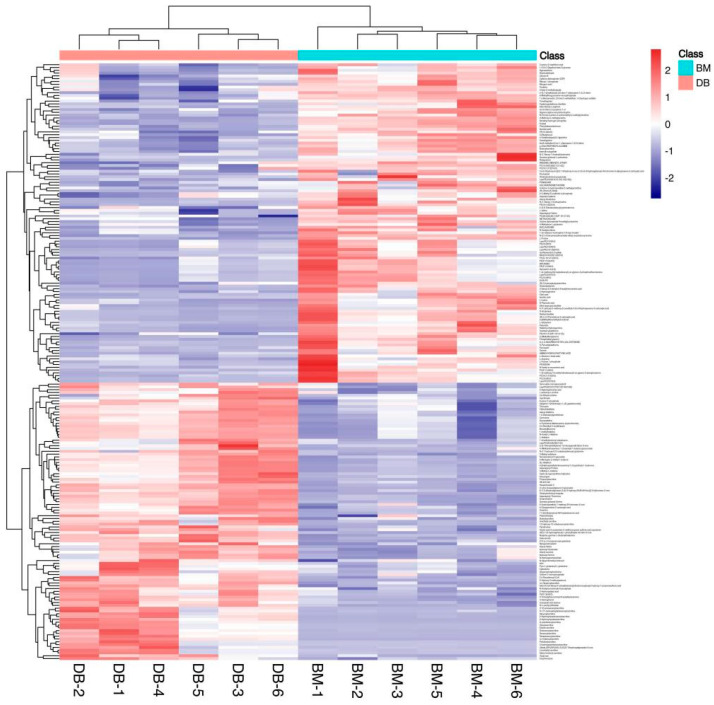
The differential metabolite heat map showed the differences between the two pigs.

**Figure 4 foods-12-00005-f004:**
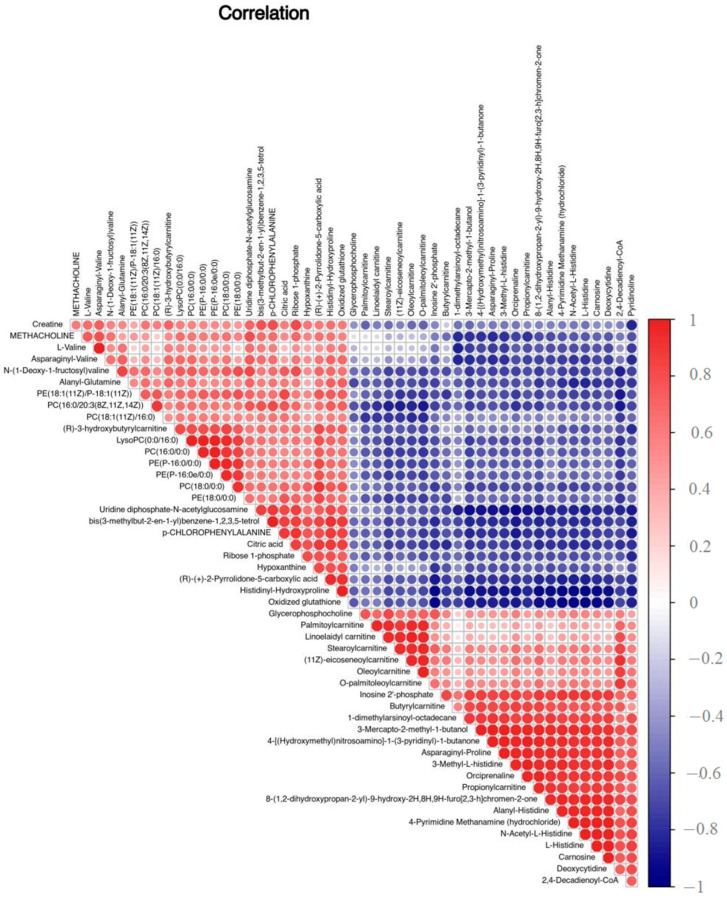
Correlation analysis chart of differential metabolites (Top50).

**Figure 5 foods-12-00005-f005:**
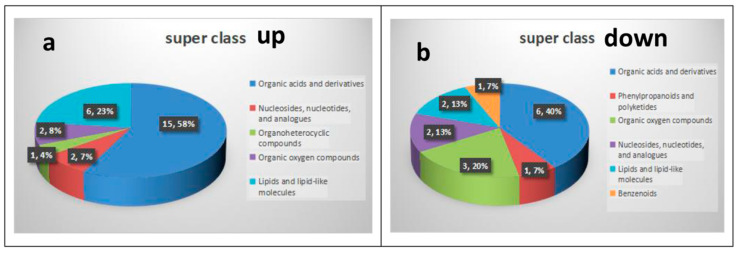
Statistical chart of upregulated (**a**) and downregulated differential metabolites (**b**).

**Figure 6 foods-12-00005-f006:**
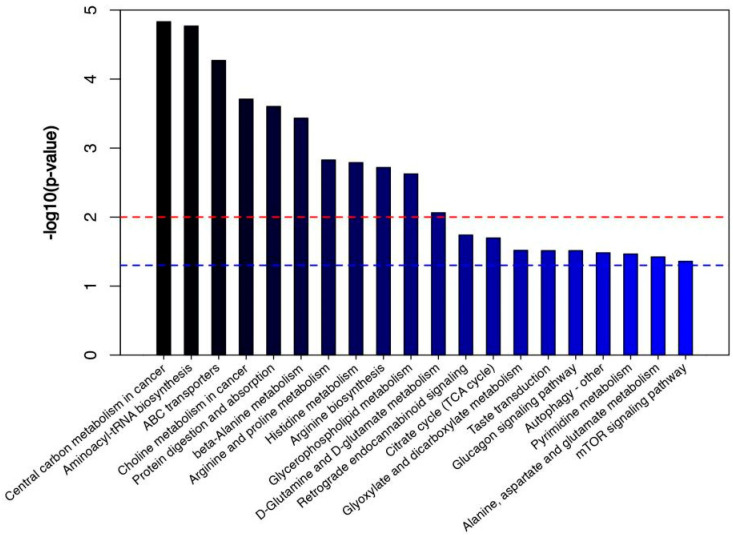
Metabolic pathway enrichment map (Top 20). The columns higher than the blue line mean *p* < 0.05, and the columns higher than the red line mean *p* < 0.01.

**Figure 7 foods-12-00005-f007:**
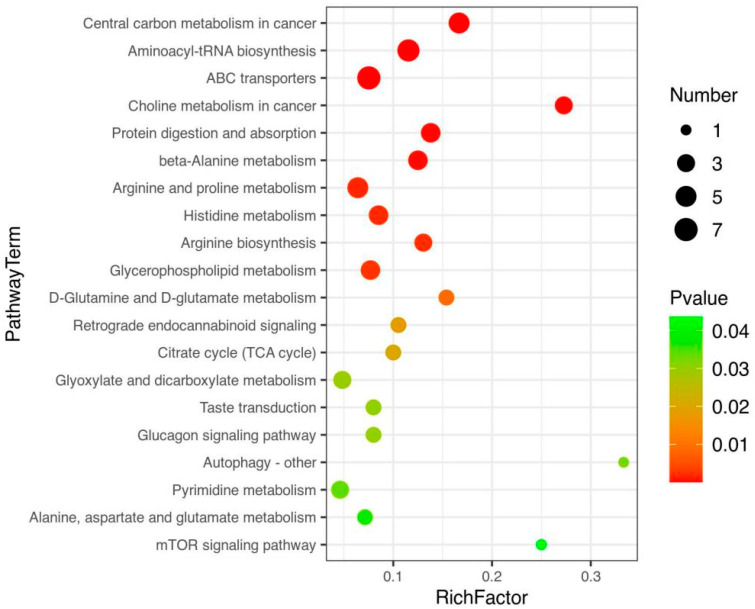
Bubble chart (using adj-P analysis).

**Figure 8 foods-12-00005-f008:**
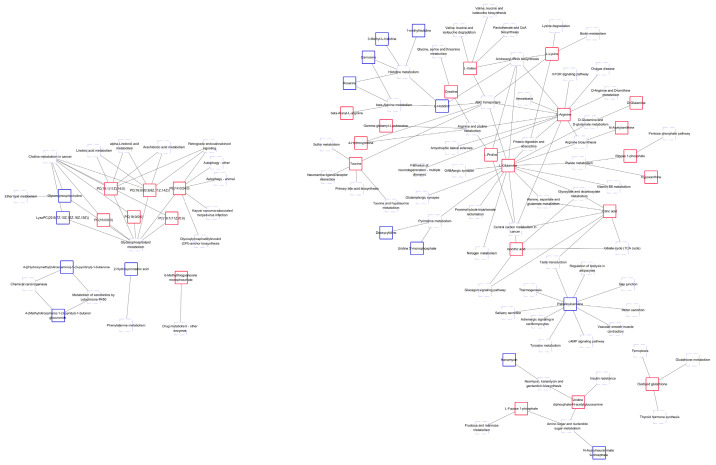
Metabolic pathways and metabolite network diagram. The red box means the metabolite with high content in BM, the green box means the metabolite with low content in BM (compared with DB), and the light-dotted boxes represent metabolic pathways. BM: Bama xiang pig, DB: Debao pig.

**Table 1 foods-12-00005-t001:** Upregulation of differential metabolites and their classification.

Metabolites	KEGG ID	Super Class
Hypoxanthine	C00262	Organoheterocyclic compounds
L-Fucose 1-phosphate	C02985	Organic oxygen compounds
Ribose 1-phosphate	C00620	Organic oxygen compounds
4-Hydroxyproline	C01157	Organic acids and derivatives
Beta-Alanyl-L-arginine	C05340	Organic acids and derivatives
Citric acid	C00158	Organic acids and derivatives
Creatine	C00300	Organic acids and derivatives
D-Glutamine	C00819	Organic acids and derivatives
Gamma-glutamyl-L-putrescine	C15699	Organic acids and derivatives
Isocitric acid	C00311	Organic acids and derivatives
L-Arginine	C00062	Organic acids and derivatives
L-Glutamine	C00064	Organic acids and derivatives
L-Lysine	C00047	Organic acids and derivatives
L-Proline	C00148	Organic acids and derivatives
L-Valine	C00183	Organic acids and derivatives
N-Acetylornithine	C00437	Organic acids and derivatives
Oxidized glutathione	C00127	Organic acids and derivatives
Taurine	C00245	Organic acids and derivatives
6-Methylthioguanosine monophosphate	C16620	Nucleosides, nucleotides, and analogues
Uridine diphosphate-N-acetylglucosamine	C00043	Nucleosides, nucleotides, and analogues
PC (18:1(11Z)/0:0)	C04230	Lipids and lipid-like molecules
PC (16:0/0:0)	C04230	Lipids and lipid-like molecules
PC (18:0/0:0)	C04230	Lipids and lipid-like molecules
PC (18:1(11Z)/16:0)	C00157	Lipids and lipid-like molecules
PC (16:0/20:3(8Z,11Z,14Z))	C00157	Lipids and lipid-like molecules
PE (14:0/24:0)	C00350	Lipids and lipid-like molecules

**Table 2 foods-12-00005-t002:** Downregulation of differential metabolites and their classification.

Metabolites	KEGG ID	Super Class
2-Hydroxycinnamic acid	C01772	Phenylpropanoids and polyketides
4-[(Hydroxymethyl)nitrosoamino]-1-(3-pyridinyl)-1-butanone	C19563	Organic oxygen compounds
Kanamycin	C01822	Organic oxygen compounds
N-Acetylneuraminate 9-phosphate	C06241	Organic oxygen compounds
3-Methyl-L-histidine	C01152	Organic acids and derivatives
1-methylhistidine	C01152	Organic acids and derivatives
4-(Methylnitrosamino)-1-(3-pyridyl)-1-butanol glucuronide	C19605	Organic acids and derivatives
Anserine	C01262	Organic acids and derivatives
Carnosine	C00386	Organic acids and derivatives
L-Histidine	C00135	Organic acids and derivatives
Deoxycytidine	C00881	Nucleosides, nucleotides, and analogues
Uridine 5′-monophosphate	C00105	Nucleosides, nucleotides, and analogues
Glycerophosphocholine	C00670	Lipids and lipid-like molecules
LysoPC (22:5(7Z,10Z,13Z,16Z,19Z))	C04230	Lipids and lipid-like molecules
Palmitoylcarnitine	C00547	Benzenoids

## Data Availability

Data is contained within the article or Appendix A.

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
