# Peer review of "Metabolomics Characterize the Differential Metabolic Markers between Bama Xiang Pig and Debao Pig to Identify Pork"

_foods, 2022, doi:10.3390/foods12010005_

Round 1
Reviewer 1 Report
The paper "foods-202627" titled "Metabolomics characterize the differential metabolic markers between Bama Xiang pig and Debao pig to identify pork Authors" by Chen and co-workers aimed to investigate the potential of metabolomics to separate between two breeds. The final objectives of this study are not clear. The experimental design used by the authors is very weak and subject to several drawbacks. First, there is no control on the animals that should be from an experimental research unit for such research instead of purchasing animals from the supermarket. Second, the feeding and productions systems should also be under control as well as the age. The animals should be reared under the same conditions to be sure that the effect that is investigated is related to the breed and not to other interacting factors. Third, we have no idea which tissue was considered in this study. No sampling time was given. No conditions of sampling are detailed nor mentioned, making serious confusion/ambiguity about this work. Fourth, there is no meat quality evaluation in this paper, making then the work not in line with Foods journal. Fifth, the statistical analyses used are not sound, and they are not considering the individual variations of the animals. The statistical model used is not clear. Sixth, the methods are weakly presented, and they are not very clear. There are serious doubts on how the data are generated, how they were handled. Seventh, the discussion is broad and not focused. The data of the paper were not scientifically discussed, and it seems that huge aspects were ignored by the authors.
Author Response
Reply:Thank you very much for your comments. These problems really exist in my manuscript, and I need to explain them to you. As follows: 1. We have controlled the variables before sampling. The materials used in this experiment are all from the same farm. Their feeding environment is the same, and their slaughter environment is the same. 2. We choose to slaughter and take samples at about 3 months old. They are the same batch of pigs, so they are the same age. 3. The tissue we selected is the longissimus dorsi muscle. I'm sorry I didn't write it in the article. I added it after modification, including sampling conditions. 4. As you said, we did not test the meat quality, but our purpose is to find differential metabolites and then distinguish pork, we did not evaluate meat quality. Meat quality related metabolites and their identification are our next step. 5. We performed secondary analysis and verification on the statistical analysis results to ensure the accuracy of the screened differential metabolites. 6. Because we don't think this part is the key of the article, we didn't give a detailed description, but wrote some important experimental conditions. We used Excel, R and an online tool for statistical analysis(https://cloud.oebiotech.com/task/ ). 7. Our discussion is about some of the top of differential metabolites. The purpose is to explain the role of these metabolites. All the differential metabolites we screened play a role. As for the data in the paper, we analyzed it in the results section, because the results we need are just to screen these differential metabolites used to distinguish different pork, which is also directly supported by the data analysis results.
In addition, we also made substantial revisions to the manuscript, including conclusions and discussions, and checked and revised the grammar of the article. All modifications have been indicated in the text
Thanks again! Perhaps our manuscript still has some shortcomings, and we hope you can give us your opinions again.
Reviewer 2 Report
The article is written using the IMRAD method with complete elements as a scientific work (original article).
The abstract section makes it easier for readers to find the entire article's contents concisely, clearly, and concisely.
In the introduction, it supports the urgency of this study, in which this study examines the differences in metabolic markers of the two pig varieties. One is the superior variety.
The method used is acceptable and up-to-date in LCMS to analyze metabolic markers comprehensively.
The data is presented in tables, pictures, and narratives that are pretty clear in language, supported by the latest literature.
Need improvement and examination related to English grammar.
Author Response
Reply: Thank you very much for your affirmation of my research. I also thank you very much for your comments, so we have made some modifications to the discussion and conclusion, as well as to grammar and other issues. Thanks again!
Reviewer 3 Report
Metabolomics-based studies, i.e. concerned with the comprehensive characterization (analysis) of metabolites and low-molecular-weight molecules in bio-fluids, tissues or whole organisms, are now of paramount relevance to the extending knowledge on influence both genetic and environmental factors. The specific study evaluates two Chinese native pig breeds, Bama Xiang (BM) and Debao black (DB), involving the M. longissimus dorsi collection and analysis of a large set of metabolic markers (molecular phenotype). Metabolic profiling yielded concentrations of as many as 213 differential metabolites of BM and DB animals (castrated males). The special importance of this paper stems from the fact that authors were able to identify 6 differential metabolites, and 4 potential differential metabolites, that may be used as endogenous biomarkers to discern meat (longest dorsal muscle tissue) of BM pig from meat of DB pig breed.
The manuscript is relatively well written with clear objectives. Background literature is well reviewed while the gap in knowledge being filled by the present work is highlighted. Methodology employed is sufficiently described. Results are presented in a comprehensible way. In the discussion section the main findings are related to those of previous studies. The relevance of these findings for further studies on pork quality and safety is also rightly stressed.
The authors should bear in mind, however, that all domestic pigs are members of a single species (Sus scrofa domestica L.). Therefore, Bama Xiang and Debao black pigs represent two different breeds (and not different species) of domestic swine (Abstract, line 11–12).
What is more, the authors have also to keep in mind that that different sections of the original research articles warrant the use of different tenses. For example, in the M&M section simple past tense should be used to describe “what you did and how you did it”. And same goes for the description of the results (Results section), where the present tense should be used only when you refer to tables, figures and graphs. This inconsistency (with the above-mentioned rules) in the submitted manuscript must be thoroughly corrected.
Author Response
Reply: First of all, thank you for your affirmation of my manuscript. We also thank you very much for your comments, which are very important, so we have revised the manuscript according to your suggestions. First of all, we modified the summary into different breeds of domestic pigs; Then we checked and revised the tenses of the full text, and we also made some modifications to the discussion and conclusion. I will pay attention to the tenses used in future research articles. Thank you very much.
Round 2
Reviewer 1 Report
I checked the paper a second time. The drawbacks of this paper are fundamental. First, the sampling time is not clear and the manner of how they were handled are not appropriate. The samples were frozen but no details of cold-shortening. Second, the number of samples used with no validation is a high risk for the quality of the paper. Third, no filetering criteria were used for the metabolites and severals of them are not exisiting in muscle. Several other drawbacks already mentionned in my previous review are major, hense making this paper of high risk in terms of scientific soundness.
Author Response
Point 1: The sampling time is not clear and the manner of how they were handled are not appropriate. The samples were frozen but no details of cold-shortening.
Response 1: We don't know which time you mean, I will list some relevant times as follows:
Slaughter time: 4:00 a.m.
Sampling time: 5:00 a.m.
Delivered to the laboratory: about 7:00 a.m.(the same day)
The frozen details has been added into the paper. During the test, the sample is placed in the - 80 ℃ refrigerator.
Point 2: The number of samples used with no validation
Response 2: We sampled six Bama Xiang pigs and six Debao pigs, one sample from each pig (the middle of the longissimus dorsi muscle), and the sampling sites were the same, a total of 12 samples.
Point 3: No filetering criteria were used for the metabolites and severals of them are not exisiting in muscle
Response3: We screened some metabolites detected, but because there were too many metabolites, they were not filtered one by one, but the available metabolites ( in abstract and results) we finally screened were all present in pig muscle. As for potential metabolites, they don't have KEGG ID,it is used as an expected alternative marker.
Point 4: Several other drawbacks already mentionned in my previous review
Response 4: We reconsidered your last review report. Our pork was bought from the slaughterhouse. Before that, we tracked all pigs, all of them from the same farm. We have tried our best to control other factors same. However, as you said, we did not buy pigs from experimental units or breed the experiment pigs ourselves, because our experiments were limited by funds and places. Hope you can understand.
Finally, except for some relatively unimportant contents, the rest are revised in the text.
Our experiment has some defects, but we think the results obtained from the experiment are still of reference value.